# Core Lab Adjudication of the ACURATE *neo2* Hemodynamic Performance Using Computed-Tomography-Corrected Left Ventricular Outflow Tract Area

**DOI:** 10.3390/jcm11206103

**Published:** 2022-10-17

**Authors:** Ahmed Elkoumy, Andreas Rück, Won-Keun Kim, Mohamed Abdel-Wahab, Mahmoud Abdelshafy, Ole De Backer, Hesham Elzomor, Christian Hengstenberg, Sameh K. Mohamed, Nawzad Saleh, Shahram Arsang-Jang, Henrik Bjursten, Andrew Simpkin, Christopher U. Meduri, Osama Soliman

**Affiliations:** 1Discipline of Cardiology, Saolta Group, Galway University Hospital, Health Service Executive and CORRIB Core Lab, University of Galway, H91 V4AY Galway, Ireland; 2Islamic Center of Cardiology, Al-Azhar University, Nasr City, Cairo 11651, Egypt; 3Department of Cardiology, Karolinska University Hospital, 17176 Stockholm, Sweden; 4Department of Cardiology and Cardiac Surgery, Kerckhoff Heart and Lung Centre, 61231 Bad Nauheim, Germany; 5Department of Cardiology, Heart Center Leipzig at University of Leipzig, 04289 Leipzig, Germany; 6The Heart Center—Rigshospitalet, 2100 Copenhagen, Denmark; 7Division of Cardiology, Department of Internal Medicine II, Medical University of Vienna, 1090 Vienna, Austria; 8CÚRAM, SFI Research Centre for Medical Devices, H91 W2TY Galway, Ireland; 9Department of Cardiothoracic Surgery, Anesthesia and Intensive Care, Skane University Hospital, Lund University, 22185 Lund, Sweden; 10Data Science Institute, University of Galway, H91 TK33 Galway, Ireland

**Keywords:** aortic stenosis, ACURATE *neo2*, left ventricular outflow tract, hemodynamic performance, computed tomography, echocardiography, prosthesis patient mismatch

## Abstract

(1) Background: Hemodynamic assessment of prosthetic heart valves using conventional 2D transthoracic Echocardiography-Doppler (2D-TTE) has limitations. Of those, left ventricular outflow tract (LVOT) area measurement is one of the major limitations of the continuity equation, which assumes a circular LVOT. (2) Methods: This study comprised 258 patients with severe aortic stenosis (AS), who were treated with the ACURATE *neo2*. The LVOT area and its dependent Doppler-derived parameters, including effective orifice area (EOA) and stroke volume (SV), in addition to their indexed values, were calculated from post-TAVI 2D-TTE. In addition, the 3D-LVOT area from pre-procedural MDCT scans was obtained and used to calculate corrected Doppler-derived parameters. The incidence rates of prosthesis patient mismatch (PPM) were compared between the 2D-TTE and MDCT-based methods (3) Results: The main results show that the 2D-TTE measured LVOT is significantly smaller than 3D-MDCT (350.4 ± 62.04 mm^2^ vs. 405.22 ± 81.32 mm^2^) (95% Credible interval (CrI) of differences: −55.15, −36.09), which resulted in smaller EOA (2.25 ± 0.59 vs. 2.58 ± 0.63 cm^2^) (Beta = −0.642 (95%CrI of differences: −0.85, −0.43), and lower SV (73.88 ± 21.41 vs. 84.47 ± 22.66 mL), (Beta = −7.29 (95% CrI: −14.45, −0.14)), respectively. PPM incidence appears more frequent with 2D-TTE- than 3D-MDCT-corrected measurements (based on the EOAi) 8.52% vs. 2.32%, respectively. In addition, significant differences regarding the EOA among the three valve sizes (S, M and L) were seen only with the MDCT, but not on 2D-TTE. (4) Conclusions: The corrected continuity equation by combining the 3D-LVOT area from MDCT with the TTE Doppler parameters might provide a more accurate assessment of hemodynamic parameters and PPM diagnosis in patients treated with TAVI. The ACURATE *neo2* THV has a large EOA and low incidence of PPM using the 3D-corrected LVOT area than on 2D-TTE. These findings need further confirmation on long-term follow-up and in other studies.

## 1. Introduction

Transcatheter aortic valve implantation (TAVI) is a recommended interventional strategy in selected patients with severe aortic stenosis (AS) [1,2]. Transcatheter heart valve (THV) hemodynamic performance requires detailed and accurate assessment at multiple time points to determine the device’s success and detect prosthetic dysfunction. The first post-TAVI hemodynamic measurements are typically obtained before or early after hospital discharge, and are considered a baseline characterization of the implanted device (fingerprint) [3,4]; further follow-up can be compared with this baseline. Transthoracic echocardiography (TTE) is the gold standard imaging modality to assess the hemodynamic performance of THVs by measuring the peak velocity, transvalvular pressure gradients, effective orifice area (EOA), stroke volume, dimensionless velocity index (DVI) and the residual post-TAVI aortic regurgitation [3,4,5,6]. Several of these hemodynamic parameters are dependent on measuring the left ventricular outflow tract area (LVOT) area.

LVOT area calculation by the monoplane 2D-TTE is based on the measurement of a single diameter in mid-systole from the parasternal long-axis view, which resembles the small anteroposterior diameter and assumes a circular shape of the LVOT. Error in area estimation due to geometric assumptions will lead to an erroneous calculation of all derived parameters such as EOA and SV [7,8]. Accordingly, the accurate assessment of hemodynamic parameters mandates reducing or eliminating errors in LVOT area measurements.

The ACURATE *neo2* (Boston Scientific, Marlborough, MA, USA), is a new generation of self-expandable, supra-annular THV, with favorable outcomes, including a significant reduction in the incidence of residual regurgitation in comparison to the earlier iteration ACURATE *neo* [9,10]. Data on the hemodynamic performance of the ACURATE *neo2* are scarce and based only on 2D-TTE.

We hypothesized that measurements of LVOT area in a 3D fashion, from gated multi-phase reconstructed multidetector computed tomography (MDCT) scans, could result in different hemodynamic performance values, and thus, reclassification of the 2D-TEE-derived ACURATE *neo2* THV outcome. In this study, we sought to assess the LVOT-dependent hemodynamic parameters such as EOA, stroke volume and their indexed values through the multimodal imaging approach, combining the more accurate 3D-MDCT-derived LVOT area and the 2D-TTE Doppler values. Reporting of these corrected values may give a true estimation of the hemodynamic performance of the ACURATE *neo2*.

## 2. Materials and Methods

This is a Core-Lab-led post hoc analysis of the Early neo2 Registry, a multi-center investigator-initiated European Registry of the first patients treated with the ACURATE *neo2* THV Prosthesis in Europe after market approval (NCT04810195). This study is a retrospective analysis of patients with severe native AS or malfunctioning bioprosthetic surgical aortic valves who underwent TAVI with the ACURATE *neo2* THV. We included patients with available pre-TAVI multi-phase MDCT scans and the comprehensive 2D-TTE assessment within seven days from the index procedure. All TTE and MDCT analyses were performed by three well-experienced senior cardiologists (AE, HE and MA).

The primary outcomes were the changes in hemodynamic classification of prosthesis patient mismatch (PPM) and the differences in the LVOT-dependent parameters (EOA, SV and their indexed values) between the 2D-TTE-derived continuity equation (CE) and the 3D-corrected CE by combining the MDCT-derived 3D-LVOT area and 2D-TTE Doppler measurements. The rate of PPM (moderate and severe PPM) between the two methods and the rate of reclassifications were reported.

### 2.1. Definition of Prosthesis Patient Mismatch (PPM)

Prosthesis patient mismatch (PPM) was identified as an EOA smaller than expected or the normal value, which led to inadequate cardiac output to meet the patient’s body demands, despite a normally functioning device without structural abnormality [2]. The indexed EOA (EOAi) is the main parameter used to assess the PPM according to the guideline’s recommendations [2,4].

For patients with BMI < 30 kg/cm^2^; PPM is:-Hemodynamically insignificant if the indexed EOA is >0.85 cm^2^/m^2^.-Moderate if between 0.66 and 0.85 cm^2^/m^2^.-Severe if ≤0.65 cm^2^/m^2^.For obese patients with BMI ≥ 30 kg/m^2^; PPM is:-Hemodynamically insignificant if the indexed EOA is >0.70 cm^2^/m^2^.-Moderate if between 0.56 and 0.70 cm^2^/m^2^.-Severe if ≤0.55 cm^2^/m^2^.

### 2.2. Echocardiography

A comprehensive 2D-TTE assessment of post-TAVI patients was performed before hospital discharge or within seven days from the index procedure according to the recommended guidelines for evaluating prosthetic heart valves [3,6]. Echocardiographic analyses were performed according to the Core Lab Standard Operating Procedures (SOP) based on the most recent guidelines recommendations [3,4,5,6], using a dedicated workstation (TOMTEC ARENA, TOMTEC Imaging Systems GmbH, Unterschlessheim, Germany). Velocity time integral (VTI) of blood flow across the THV (VTI_AV_) was measured from the Continuous-wave Doppler (CWD) and that of the LVOT (VTI_LVOT_) was measured from the pulsed-wave Doppler (PWD) of LVOT. Both measurements were obtained from the 3- or the 5-chamber apical views, if appropriate. The sample volume for the VTI_LVOT_ was typically positioned at the LV edge of the THV in systole. As recommended by guidelines, the external LVOT diameter was measured from the parasternal long-axis view in zoomed view below the prosthetic stent (inflow level) in mid-systole (Figure 1). LVOT area was calculated automatically with the formula (A = πr^2^) and used to calculate the EOA using the CE in addition to the calculation of SV across the LVOT using the flow equation (Flow_LVOT_ (SV_LVOT_) = LVOT area × VTI_LVOT_). All hemodynamic parameters values were indexed to the patient’s body surface area.

### 2.3. Multidetector Computerized Tomography (MDCT)

Pre-TAVI MDCT scans acquisition was performed according to each center’s protocol. Offline 3D multiplanar reconstruction and comprehensive analysis were performed according to the Core Lab SOP in accordance with the Society of Cardiac Computed Tomography (SCCT) guidelines [11]. The LVOT was measured at 5 mm below and perpendicular to the predefined native aortic annulus level from contrast enhanced MDCT scans, using a dedicated workstation (3mensio^®^ Structural Heart 10.2, 3mensio Medical Imaging, B.V., The Netherlands). Direct planimetry of the LVOT area and the diameters were measured in the enface and zoomed view as vertical (Minimum = Dmin) and horizontal (Maximum = Dmax) on the mid-late systole (30–40% systolic phases) (Figure 1). The diameters were used to calculate the LVOT eccentricity index. The eccentricity index was calculated to define the shape of the LVOT (circular or elliptical) using the formula [1 − (Dmin/Dmax)] × 100. LVOT is considered circular when the eccentricity index was <10% [12].

### 2.4. Corrected Continuity and Flow Equations

The EOA and SV were calculated using the conventional 2D-TTE-derived parameters; post-TAVI EOA _TTE_ = [(LVOT area _TTE_ × PWD VTI_LVOT_)/CWD VTI_AV_]. Post-TAVI SV (SV_TTE_) was calculated as SV_TTE_ = [(LVOT area _TTE_ × PWD VTI _LVOT_).

On the other hand, the corrected equations indicate the use of the MDCT-derived 3D-LVOT area (Direct planimetry from MPR views without geometric assumptions) to be used in the calculation of AV EOA and SV instead of the TTE-derived LVOT area (based on the assumption of circular LVOT shape). Therefore, the corrected parameters were calculated as follows.
Post-TAVI EOA_MSCT_ = [(LVOT area _MDCT_ × PWD VTI_LVOT_)/CWD VTI _AV_]

Post-TAVI SV_MSCT_ = [(LVOT area _MDCT_ × PWD VTI _LVOT_). In addition, EOA and SV were indexed to patients’ BSA to calculate the indexed corrected parameters.

We compared the PPM rate between the 2D-TTE and the 3D-Corrected-MDCT CE-derived EOAi values.

### 2.5. Statistical Analysis

Results are presented as mean ± standard deviation (SD) or median with interquartile range (IQR), according to their distribution pattern. We used the Shapiro–Wilks test as well as QQ plot to assess the normality of continuous variables. Categorical data were presented as percentages and fractions of occurrence. Correlation and agreement between the LVOT area and LVOT area dependent parameters, obtained by different methods (2D TTE and MDCT), were determined using Pearson correlation, Spearman rank correlation and Bland–Altman analysis, respectively. Correlation and agreement between mean trans prosthetic PG, with the EOA and EOAi, calculated from TTE and MDCT. 

Intra-observer and inter-observer (two independent blinded observers) reproducibility of LVOT area measured by TTE and MSCT was performed in a random set of 20 patients and evaluated using the intraclass correlation coefficient for absolute agreement. Good agreement was defined as >0.80. Mean transprosthetic PG was scatter-plotted for each imaging-technique-derived EOA and EOAi and fitted curves for data pairs were constructed.

The Bayesian mixed-effect model was used to account the cluster effects of measurements, while parameters obtained from two methods are nested within patients.

Bayesian mixed-effect models with gaussian and asymptotic Laplace priors based on the distribution pattern of the dependent variables were used to compare the quantitative parameters between the two groups. While gaussian and asymptotic Laplace were used for normal and skewed distribution, respectively. Furthermore, Bayesian mixed-effect models with Bernoulli (binary) and cumulative priors (ordinal) were used to compare the PPM rate between the two methods. We also used the mixed-effect Bayesian regression model to compare changes in the hemodynamic performance of ACURATE *neo2* among small (23 mm), intermediate (25 mm) and large (27 mm) sizes after implantation.

The convergence of the Bayesian models was examined using R-hat, LOO, and posterior predictive plots. The R-hat < 1.1 indicates a suitable model of convergence. All statistical analyses were conducted using the ggplot2 and rstan packages in the R 4.1.1 environment.

The posterior Beta or Odds ratio (OR) was used to report the associations between variables of interest. The 95% credible interval (Crl) was used to examine the differences between the two groups, Crl includes zero value for continuous models and one for categorical models indicating non-significant associations.

## 3. Results

A total of 554 patients with severe AS were treated with TAVI using ACURATE *neo2* between September 2020 and April 2021 and included in the Early neo2 Registry. We excluded patients with MDCT without mid–late systolic phases, patients who required valve in valve bailout therapy with a device other than ACURATE *neo2*, and patients without either post-TAVI 2D TTE study or pre-TAVI MDCT available in the Core Lab for the independent analysis. In total, 258 patients comprised the final cohort of this study. Mean age was 81.6 ± 6.1 years, 65% women with a median of the European System for Cardiac Operative Risk Evaluation (EuroSCORE) II of 3.34% [2.15, 3.5]. The baseline characteristics of the study population are shown in Table 1. The median duration between the pre-TAVI MDCT scan and the TAVI procedure was 13 days [2, 46]. The study cohort included eight patients (3.1%) with type I bicuspid AV, and six patients with TAVI in malfunctioning surgical AV prosthesis (TAVI in SAVR). Pre-TAVI MDCT scans analysis revealed a mean of the native annulus area of 430.2 ± 62.9 mm^2^, LVOT minimum and maximum diameters of 19.03 ± 2.55 mm, and 26.92 ± 2.43 mm, respectively with a measured LVOT area of 405.22 ± 81.32 mm^2^, and LVOT eccentricity index 29.21 ± 7.4% indicating that LVOT area was oval (Elliptical) in 257 (99.5%) cases (Table 1). All patients were treated via transfemoral vascular access, balloon pre-dilatation was performed in 81.8%, while post-dilatation was performed in 41.1% (Table 2).

### 3.1. Hemodynamic Outcomes (Conventional 2D TTE and MSCT-Corrected Parameters)

Post-procedural 2D-TTE assessment revealed LVEF of 58.9 ± 9.8%, AV maximum velocity 1.98 ± 0.44 m/s, trans-prosthetic mean pressure gradient 7.22 ± 3.11 mmHg and dimensionless velocity index (DVI 0.64 ± 0.13. Post-TAVI residual AR assessment revealed 59.7% of patients with none/trace AR, 36.4% had mild AR, 1.9% with moderate AR and none had severe AR (Table 3).

The mean LVOT diameter on 2D-TTE was 21.03 ± 1.9 mm and shows a significant difference between the LVOT dimensions obtained from the MDCT scan, Dmin 19.03 ± 2.55 mm (95% Crl of differences: 1.7, 2.31) and Dmax 26.92 ± 2.43 mm (95% Crl of differences: −6.2, −5.58).

The mean LVOT area obtained from TTE and MDCT were 350.4 ± 62.04 mm^2^ and 405.22 ± 81.32 mm^2^, respectively (95% CrI of differences: −55.15, −36.09), which resulted in a smaller EOA and lower SV (2.25 ± 0.59 vs. 2.58 ± 0.63 cm^2^) and (73.88 ± 21.41 vs. 84.47 ± 22.66 mL), (Beta = −0.642 (95%CrI of differences: −0.85, −0.43), (Beta = −7.29 (95% CrI: −14.45, −0.14)), respectively and consequently the indexed values (EOAi 1.20 ± 0.32 cm^2^/m^2^ vs. 1.41 ± 0.34 cm^2^/m^2^ (95% CrI of differences: −0.207, −0.136), SVi TTE 41 ± 12.6 mL/m^2^ vs. 46.14 ± 12 mL/m^2^ (95% CrI of differences: −0.207, −0.136) (Table 4).

### 3.2. Prosthesis-Patient Mismatch (PPM) Incidence and Reclassification

The incidence of all (overall) PPM measured by conventional 2D-TTE (8.52%) was higher than MDCT-corrected formula (2.32%), OR = 8.36 (95% Crl: 2.42, 39.61), (Kappa w = 0.323, 95% confidence interval (CI): 0.13, 0.51). However, the differences remained statistically significant in the adjusted model by sex, age, and BMI variables (OR = 10.33; 95% CrI: 2.5, 67.34). The distributions of PPM frequency within BMI categories are shown in Figure 2.

### 3.3. Stroke Volume Index Changes in Patients with Low EF%

In 29 patients with EF < 50% (mean of EF was 40.47 ± 6.49%), the SVi changed significantly from 34.1 ± 11.4 mL/m^2^ by TTE to 39.3 ± 11 mL/m^2^ with MDCT LVOT-corrected calculation (Beta = 5.18; 95% CrI: 2.36, 8).

### 3.4. Inter Valve Size Differences in Hemodynamic Performance and Incidence of PPM

According to the results of the mixed-effects model adjusted for age, sex, BMI, and BSA, to determine the effect of other variables on MDCT and TTE, the detection ability of the interaction effects between methods and independent variables was tested (Appendix A).

The EOA*ACURATE *neo2* size interaction was statistically significant; thus, subgroup analysis according to ACURATE *neo2* sizes indicated that mean differences in EOA between TTE and MDCT were obvious for the 23 mm (diff = 0.64, 95% CrI: 0.44, 0.85) compared with the 25 mm (diff = 0.208, 95% CrI: 0.03, 0.35) and 27 mm (diff = 0.26, 95% CrI: 0.11 0.44). The interaction effects between methods and the rest of the independent variables were insignificant. (Appendix A).

In the simple Bayesian logistic regression model, a higher risk of PPM was observed for ACURATE *neo2* size 23 mm than ACURATE *neo2* size 25 and 27 mm (OR = 3.57; 95% CrI: 1.12, 12.2) with 2D-TTE. With the MDCT, there was no association between the size of ACURATE *neo2* and PPM (Appendix A).

### 3.5. Intra-Observer and Inter-Observer Reliability

An excellent agreement was observed for the intra-observer and Inter observer reliability regarding LVOT area measured by MDCT (ICC = 0.99 [95% CI; 0.98–0.99]) and (ICC = 0.98 [95% CI; 0.95 to 0.99]), respectively and was good regarding TTE (ICC = 0.87 [95% CI; 0.71, 0.95]) and (ICC = 0.85 [95% CI; 0.63, 0.94]) (Appendix A).

## 4. Discussion

This is the first study that systematically evaluates the new supra-annular ACURATE *neo2* THV hemodynamic performance using the LVOT area derived from both the conventional 2D-TTE and pre-procedural MDCT scan 3D measurements. Both techniques were used to calculate all LVOT-dependent hemodynamic parameters (EOA, SV in addition to their indexed values) aiming to accurately report hemodynamic parameters outcome early at patients’ hospital discharge and to define the baseline hemodynamic performance of further follow up in comparison to the obtained values, especially for the diagnosis and severity of PPM.

The main findings of this report are as follows; (1) the calculation of LVOT area from 2D-TTE significantly underestimated the area in comparison to the MDCT measured 3D-LVOT area (350 vs. 405 mm^2^), and all LVOT-dependent parameters; (2) furthermore, the LVOT was oval in most cases (99.5%) with a mean eccentricity index of 29.2%; (3) recalculation of EOAi resulted in a significant reduction in PPM incidence among the included cohort (8.5% to 2.3%), (4) 3D-MDCT-corrected LVOT are measurements resulted in obtaining more concordance between EOA and other hemodynamic parameters; and (5) finally, the results also show a significant difference between the different sizes with the use of the corrected LVOT assessment in contrast to the conventional TTE assessment.

The fact of measuring a 3D structure using a 2D image usually carries the risk of inaccurate assessment. In LVOT area measurements, our results confirm significant underestimation of the LVOT area in agreement with multiple reports. Liu et al. compared LVOT area measurements using biplane versus single dimensions using TTE, and resulted in the LVOT with the biplane method being larger than the conventional method (420 vs. 373 mm^2^) [13]; in addition, Weber et al., have reported that the MDCT-derived LVOT area was larger than 2D-TTE (456.9 vs. 303.7 mm^2^) and resulted in larger EOA in patients with severe AS and reclassification of 30% of the included cohort from severe to moderate AS [14].

The concept of using accurately measured LVOT (3D-LVOT) to be included in the CE is a quite old seeking more accurate and reproducible results [15,16,17], but the application and the use of 3D-LVOT area (3D echocardiography, MDCT, or CMR) area combined with TTE Doppler (CWD and PWD) to obtain AV EOA and SV (corrected parameters) still uncommon practice. However, it could be used especially if discordance in the parameters was noticed either pre- or post-AV replacement or when PPM is suspected [15,16,17]. Multiple reports confirm the utility of the corrected calculation of EOA using the LVOT area measured from MDCT scan either pre- or post-AV replacement or even for the prediction of EOA, and mainly for the diagnosis of PPM [12,14,18]. The incidence of overall and/or severe PPM after TAVI was reported to be lower than SAVR [19,20], especially with self-expandable, supra-annular devices with larger EOA and lower gradients [21].

The incidence of PPM according to the MDCT-corrected EOAi resulted in a lower frequency of all PPM and BMI-adjusted PPM than 2D-TTE (8.5% vs. 2.3%). (Figure 2 and Table 4) the results agree with those of Fukui et al., who reported larger EOAi (1.57 vs. 1.1 cm^2^/m^2^) and reclassification of all PPM from 19.5% by TTE to 3.5% with MDCT 3D-LVOT correction for both SEV and BEV [12]. As larger devices are expected to provide larger EOA, a sub-analysis has been performed according to the implanted ACURATE *neo2* size, revealed a significant difference in EOA between the medium, 25 mm, and large, 27 mm, sizes in comparison with the small, 23 mm, devices when the MDCT-corrected LVOT area was used instead of the 2D-TTE LVOT area, which showed a non-significant difference (Appendix A).

This study recommends that LVOT area should be directly measured on a 3D imaging modality such as MDCT in all cases, if possible. Correlation of this study findings of misclassified PPM cases might offer an explanation on the lack of clinical correlation of PPM following TAVI in earlier publications. Those cases were most probably misclassified due to underestimation of LVOT area, and consequently EOA.

### Study Limitations

Although this is the first study to provide comprehensive hemodynamic reassessment and describes the recalculation of EOA and SV after implantation of the ACURATE *neo2* THV using the MDCT-derived LVOT area, some limitations exist. First, this is a retrospective study with small sample size. Second, long-term clinical outcomes of the PPM reclassification between the two methods are not available. Third, we used the same cutoff values established for the TTE assessment. Therefore, new cut-off values of PPM based on 3D-derived EOA should be derived from long-term outcome studies, and finally, the use of the pre-TAVI MDCT to measure the 3D-LVOT area, but we thought that with the short time interval between the pre-procedural MDCT and the TAVI procedure (13 days) and the post-procedural TTE, no significant changes in the LV mass will occur. Additionally, the low radial force of the ACURATE *neo2* will not significantly affect the shape of the LVOT.

## 5. Conclusions

LVOT is eccentric in most patients undergoing TAVI, which might lead to erroneous estimation of hemodynamic performance of THV from 2D-TTE using the continuity equation. Using the directly measured LVOT area on a 3D MDCT scan, instead of 2D-TTE, in combination with the TTE Doppler might reduce these limitations, and could result in an accurate and reproducible assessment of continuity-equation-derived parameters. The correction of the LVOT area showed a lower rate of PPM diagnosis dependent on the EOAi, resulting in a better correlation with other hemodynamic parameters, such as mean gradient. Using the MDCT-corrected measurements, the ACURATE *neo2* THV, a self-expandable supra-annular valve, provides a very low rate of PPM, a large EOA associated with a low trans-prosthetic pressure gradient.

## Figures and Tables

**Figure 1 jcm-11-06103-f001:**
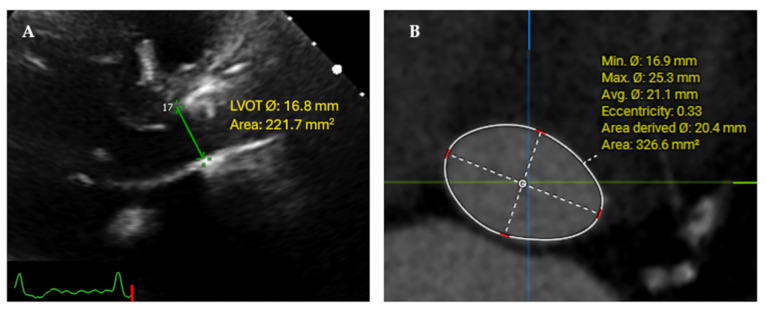
LVOT measurements: (**A**) 2D-TTE LVOT diameter measurements from the parasternal long axis zoomed view in mid-systole, (**B**) MDCT multiplanar reconstruction of the LVOT (5 mm below the annular plane, in mid-systolic phase 30%) with the minimum (anteroposterior) diameter and the maximum (medio-lateral) diameter with measured 3D-LVOT area, with a larger area calculated from the MDCT measured minimum and maximum diameters with eccentricity index of 0.33.

**Figure 2 jcm-11-06103-f002:**
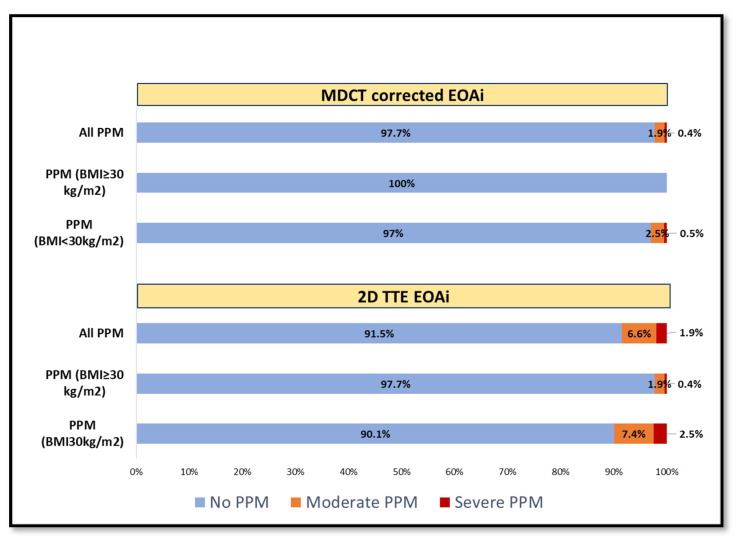
Incidence of PPM (BMI adjusted) based on the EOAi assessed by 2D-TTE and MDCT-corrected method; (All PPM; moderate or severe (2.3% (MDCT-corrected) vs. 8.5% (2D-TTE).

**Table 1 jcm-11-06103-t001:** Patients’ baseline characteristics, pre-procedural Echocardiography and MDCT scan.

Characteristic	*n* = 258
Age	81.6 (6.1)
Women	168 (65%)
Body surface area, m^2^	1.8 [1.7–2.0]
Body mass index, kg/m^2^ Body mass index < 30 kg/m^2^ Body mass index ≥ 30 kg/m^2^	26 [23.7–29.3] 204 (79%) 54 (21%)
Euroscore II, %	3.34 [2.15–3.5]
Hypertension	212 (82.2%)
Diabetes mellitus Type I Diabetes mellitus Type II	32 (12.4%) 49 (19%)
Baseline creatinine, mg/dL	1.0 [0.8–1.3]
Prior Atrial fibrillation	102 (39.5%)
Chronic lung obstructive disease	39 (15.1%)
Prior stroke or TIA	33 (12.8%)
Peripheral arterial disease	30 (11.6)
Prior permanent pacemaker implantation	26 (10.1%)
Previous cardiac surgery	30 (11.6%)
Previous CABG	15 (5.8%)
Previous PCI	51 (19.8)
New York Heart Association (NYHA) class	
Class II Class III Class IV	86 (33.3%) 141 (54.7%) 25 (9.7%)
Valve-in-Valve procedure (TAVI-in-SAVR)	6 (2.3%)
**Preprocedural 2D-TTE characteristics**	
LV Ejection fraction, %	60 [55–65]
Aortic valve maximum velocity, m/s	4.29 (0.56)
Mean pressure gradient, mmHg	43.6 [35–52]
Aortic valve effective orifice area, cm^2^	0.7 [0.6–0.8]
Moderate-severe aortic regurgitation	28 (10.9%)
Moderate-severe mitral regurgitation	38 (14.8%)
Moderate-severe tricuspid regurgitation	21 (8.2%)
**Pre-procedural MDCT characteristics**	
Bicuspid Aortic Valve (Type I)	8 (3.1%)
Native aortic annulus area, mm^2^	430.2 (62.9)
MDCT-derived LVOT measurements	
Minimum diameter, mm Maximum diameter, mm LVOT area, mm^2^	19.03 (2.55) 26.92 (2.43) 405.22 (81.32)

Values are either Median [IQR], Mean (±SD), and *n* (%).

**Table 2 jcm-11-06103-t002:** Procedural characteristics and In-hospital outcomes.

	*n* (%)
Vascular access	
Transfemoral	258 (100%)
Balloon pre-dilatation	211 (81.8%)
ACURATEneo2 size	
Small {23 mm} Medium {25 mm} Large {27 mm}	59 (22.9%) 101 (39.1%) 98 (38%)
Balloon post-dilatation	106 (41.1%)
Valve embolization	1 (0.4%)
Need for second valve implantation	1 (0.4%)
Annular injury (rupture)	0
Cardiac tamponade	0
Procedural death	0
Coronary obstruction	0
New postoperative permanent pacemaker	18 (7%)
Major vascular complications	4 (1.6%)
Major bleeding	4 (1.6%)
Life-threatening bleeding	3 (1.2%)
In-hospital stroke	7 (2.7%)
Conversion to surgery	0
New dialysis	0
All-cause mortality	0

Values are presented as *n* (%).

**Table 3 jcm-11-06103-t003:** Post-procedural TTE-Doppler assessment.

	TTE (*n* = 258)
LV ejection fraction, %	58.9 (9.8)
AV maximum velocity, m/s	1.98 (0.44)
AV mean pressure gradient, mmHg	7.22 (3.11)
Dimensionless velocity index	0.64 (0.13)
Systolic pulmonary artery pressure, mmHg	36.8 [29.5–44.1]
Post-TAVI aortic regurgitation	
None/trace Mild Moderate	154 (59.7%) 94 (36.4%) 5 (1.9%)
Moderate–severe mitral regurgitation	36 (15.2%)
Moderate–severe tricuspid regurgitation	49 (24.5%)

Values are either Median [IQR], Mean [±SD] and *n* (%).

**Table 4 jcm-11-06103-t004:** LVOT-dependent hemodynamic parameters (TTE- vs. MDCT-derived LVOT area).

	TTE	MSCT	95% CrI of Difference
LVOT diameter, mm	21.03 (1.9)	Minimum diameter 19.03 (2.55)	[1.7, 2.31]
		Maximum diameter 26.92 (2.43)	[−6.2, −5.58]
LVOT area, mm^2^	350.4 (62.04)	405.22 (81.32)	[−55.15, −36.09]
EOA, cm^2^	2.25 (0.59)	2.58 (0.63)	[−0.85, −0.43]
EOA index, cm^2^/m^2^	1.20 (0.32)	1.41 (0.34)	[−0.207, −0.136]
LVOT SV, mL	73.88 (21.41)	84.47 (22.66)	[−14.45, −0.14]
LVOT SV index, mL/m^2^	41.0 (12.6)	46.14 (12)	[−0.207, −0.136]
Prosthesis-Patient Mismatch (PPM) - All PPM - BMI adjusted PPM	22 (8.52%)	6 (2.32%)	
BMI < 30 kg/m^2^			
Moderate PPM	15 (7.3%)	5 (2.5%)	
Severe PPM	5 (2.5%)	1 (0.5%)	
BMI ≥ 30 kg/m^2^			
Moderate PPM	2 (3.7%)	0	
Severe PPM	0	0	

Values are either Median [IQR], Mean (±SD) and *n* (%); BMI = Body Mass Index.

## Data Availability

All data are available within the abstract.

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
