# Peer review of "Core Lab Adjudication of the ACURATE neo2 Hemodynamic Performance Using Computed-Tomography-Corrected Left Ventricular Outflow Tract Area"

_jcm, 2022, doi:10.3390/jcm11206103_

Round 1
Reviewer 1 Report
Elkoumy and colleagues, in their article Core Lab Adjudication of The ACURATE neo2 Hemodynamic Performance Using Computed Tomography Corrected Left Ventricular Tract Area”, describe a more accurate hemodynamic calculation and, in conclusion, reduced PPM with 3D-LVOT measurement on MDCT in patients who received an ACURATE neo2 transcatheter valve.
For their study, the authors chose a retrospective, multicenter approach, which of course reduces power but is sufficient to answer the question. The study design is adequate.
The article has some minor weaknesses that should be addressed:
- The repeated pronunciation of "new ACURATE neo2" (e.g., line 347) gives the impression that this is an industry-sponsored study. If this is the case, this needs to be highlighted and also listed in the limitations.
- Lines 347 to 451 should be deleted. These results have nothing to do with the question and topic of the article.
- The reference to 3D TTE/TEE (e.g., line 353) for better LVOT measurement is not described here and should therefore be deleted.
Author Response
Thank you very much for your review and valuable comments that helped us to improve the submitted manuscript.
- The repeated pronunciation of "new ACURATE neo2" (e.g., line 347) gives the impression that this is an industry-sponsored study. If this is the case, this needs to be highlighted and also listed in the limitations.
# Response: thank you very much for this note, regarding this point we confirm that this study is conducted fully independent from industry and the publication fees is also not covered by industry
In fact, we stated the source of funding in the paper as “Funding: “This research received support from Science Foundation of Ireland (Grant # 13/RC/2073_P2).” (Line-398).
- Lines 347 to 451 should be deleted. These results have nothing to do with the question and topic of the article.
# Response: Thank you very much for your suggestion. We removed this paragraph as suggested.
- The reference to 3D TTE/TEE (e.g., line 353) for better LVOT measurement is not described here and should therefore be deleted.
# Response: Thank you very much for your comment. We have rephrased the paragraph as suggested.
Reviewer 2 Report
This is a retrospective observational study focusing on the hemodynamic assessment of the Accurate NEO2 valve. In this study, the post-procedural area (EOA) and stroke volume (SV), were calculated from post-TAVI 2D, together with a hybrid continuity equation (with the LVOT calculated based on pre-procedural MDCT scans). The findings are interesting and in concordance with previous similar studies. The methods are clear, the results accurate and the discussion appropriate.
Major concerns:
This study has been designated and conducted based on a big assumption. The assumption was that there was no significant change in the LVOT size and shape after the TAVI. This is the most important limitation of the study. A better design of the study would command MDCT assessment of the LVOT after TAVI.
Minor issues:
Minor comments that need to be addressed to improve the understanding and readability of the manuscript.
· Abstract: line 45. please provide PPM in full words
· Line 74. Please add ….in mid-systole….
· Line 102. Please provide here a definition of PPM
· Line 152. What do you mean by ‘’ Direct planimetry from MPR views without geometric assumptions’’? Do you mention that because we usually do the assumption of a circular LVOT based on the TTE measurement?
· Line 311. …In most cases…. please add in brackets the (99.5% of the cases)
· Line 319. Reportà reports
· Did you notice any significant discrepancy between estimated EOAs and SVs in patients with Bicuspid Aortic Valve? What was the rationale for proceeding with the Accurate Neo2 in these patients with BAV and not choosing and Balloon expandable valve or a SEV with greater radial force?
Author Response
Thank you very much for your review and valuable comments that helped us to improve the submitted manuscript.
- Major concerns: This study has been designated and conducted based on a big assumption. The assumption was that there was no significant change in the LVOT size and shape after the TAVI. This is the most important limitation of the study. A better design of the study would command MDCT assessment of the LVOT after TAVI.
# Response: Thank you very much for your valuable comment. We agree with this reviewer that an ideal study design would have included post TAVI MDCT scan. Therefore, we acknowledged this design issue in the limitation section. However, our assumptions are based on three main considerations
- The very short duration between the Pre-TAVI MDCT scan and the index TAVI procedure (median duration was 13 days) precludes significant remodelling of the LVOT.
- The MDCT-derived LVOT area was combined with the 2D-TTE in the early post peri-discharge assessment (within 7 days from the index procedure).
- ACURATE noe2 device has a low radial force, so no significant change in the shape of the LVOT is expected.
Accordingly, based on these considerations non-significant changes within the LVOT geometry will be affecting the LVOT measurements with such short time frame.
Regarding the post TAVI MDCT, in Europe it is still not routine for all patients, and only required if there is a clinical indication like suspected valve thrombosis or complicated endocarditis.
- Minor issues: Minor comments that need to be addressed to improve the understanding and readability of the manuscript.
- Abstract: line 45. please provide PPM in full words
# Done
- Line 74. Please add ….in mid-systole….
#Done
- Line 102. Please provide here a definition of PPM
#Done we change its position in the manuscript from line 176 to be in line 103
- Line 152. What do you mean by ‘’Direct planimetry from MPR views without geometric assumptions’’? Do you mention that because we usually do the assumption of a circular LVOT based on the TTE measurement?
# Yes, we just confirming the superiority of MDCT in elimination of the shape assumption as a major limitation of the 2D-TTE.
- Line 311. …In most cases…. please add in brackets the (99.5% of the cases)
#Done
- Line 319. Reportà reports
#Done
- Did you notice any significant discrepancy between estimated EOAs and SVs in patients with Bicuspid Aortic Valve? What was the rationale for proceeding with the Accurate Neo2 in these patients with BAV and not choosing and Balloon expandable valve or a SEV with greater radial force?
# Response: Thank you for raising this question, unfortunately we didn’t perform a subgroup analysis of BAV as the study included a small number of only 8 patients with type 1-a BAV.
Regarding the device selection, all patients were included in the Early Neo2 registry, and the device selection was the decision of the local heart team in each participating site.